# Probabilistic classification of late treatment failure in uncomplicated falciparum malaria

Somya Mehra [1,2] ✉, Aimee R. Taylor [3], Mallika Imwong [1,4], Nicholas J. White [1,2] & James A. Watson [2,5]

Distinguishing treatment failure (recrudescence) from reinfection in uncomplicated falciparum malaria is essential for characterising antimalarial treatment efficacy in malaria endemic areas. Classification of recrudescence versus reinfection is usually based on a comparison of parasite allelic calls derived from PCR amplification and electrophoresis of individual polymorphic markers in the acute and recurrent blood samples. Match-counting methods (e.g., 3/3 or 2/3 matching alleles) have usually been applied, but these do not account for multiple comparisons per-marker when infections are polyclonal. We show that when infections are polyclonal, as is common in high transmission settings, currently used match-counting and model-based methods may have unacceptably high false-discovery rates leading to overestimation of treatment failure. We develop the software PfRecur which provides analytical Bayesian posterior probabilities of treatment failure in recurrent falciparum malaria. We use data from a recent study in Angola to demonstrate the potential utility of our model in resolving complex polyclonal *P. falciparum* infections, thereby providing more accurate estimation of treatment failure rates.

*Plasmodium falciparum* is estimated to cause around 250 million symptomatic malaria cases each year, the majority of which are in young children in sub-Saharan Africa[1]. The primary therapeutic goal of antimalarial treatment for uncomplicated malaria is to cure the infection. Effective treatment of uncomplicated malaria currently relies on artemisinin-based combination therapies (ACTs). The ACTs combine a rapidly acting but rapidly eliminated artemisinin derivative with a less active and slowly eliminated partner drug (for example, the combination of artemether and lumefantrine, AL). Artemisinin resistance is now widespread in Southeast Asia[2,3] and recently has emerged independently in East Africa[4,5]. Resistance to artemisinin results in an increased likelihood that the ACT fails to clear the blood stage infection and recrudesces. This amplifies the selective advantage of artemisinin resistant parasites and augments selection of partner drug resistance. Characterising ACT failure rates is essential to guide

policies and practices. This assessment is routinely performed through therapeutic efficacy surveillance studies. These are usually single arm observational studies which enrol symptomatic uncomplicated malaria patients and follow them for one to two months after treatment. The objective is to estimate the proportion of patients who do not clear their infection[6,7]. Treatment failure rates should not exceed 10%[1]. In areas of high transmission, where the main burden of disease is in young children and where reinfection within one month is very common, estimating the antimalarial treatment failure rate relies on distinguishing recurrent bloodstream infections resulting from incomplete clearance of the incident infection (recrudescence) from new infections (reinfection) resulting from new mosquito bites[8,9]. This relies on molecular correction methods which compare parasite genotypes of the baseline infection with those of the recurrent infection.

[1]Mahidol Oxford Tropical Medicine Research Unit, Faculty of Tropical Medicine, Mahidol University, Bangkok, Thailand. [2]Centre for Tropical Medicine and Global Health, Nuffield Department of Medicine, University of Oxford, Oxford, UK. [3]Infectious Disease Epidemiology and Analytics G5 Unit, Institut Pasteur, Université Paris Cité, Paris, France. [4]Department of Molecular Tropical Medicine and Genetics, Faculty of Tropical Medicine, Mahidol University, Bangkok, Thailand. [5]Infectious Diseases Data Observatory, Big Data Institute, Oxford, UK. ✉e-mail: somya@tropmedres.ac

Classification of a recurrent infection as either a reinfection or a recrudescence is done primarily by PCR genotyping. Length polymorphisms within genes or microsatellite markers are compared in paired acute and recurrent infection blood samples[9]. Classification is then based on the observed alleles. Most studies have used match-counting approaches, such as those recommended by the World Health Organisation (WHO)[9]. Match-counting algorithms usually take the presence of one or more shared alleles per marker in the paired samples as evidence of recrudescence, and can either be strict (requiring matches at all markers) or relaxed (requiring matches at a subset of markers). However, match-counting does not account for multiple comparisons when there are multiclonal samples (infections caused by multiple parasite clones), the allele specific sensitivity of the genotyping method[10], the relative frequency of the different alleles, and the uncertainty in the allocation of alleles across individual haploid parasite clones within each sample[8,11]. Although deep sequencing of longer amplicons from SNP-rich genomic regions (AmpSeq) can address some concerns, it is not yet widely available, and it remains necessary to use statistical model-based approaches. Model-based approaches for molecular correction can address classification uncertainty, take into account background allele frequencies, and adjust for multiple comparisons. In our view, additional biological complexities (for example, the presence of an asynchronous sequestered clone, or amplification of residual gametocyte DNA post-treatment[12]) constitute a distinct problem. These issues are beyond the scope of a general-purpose statistical model for molecular correction.

The US CDC has developed a Bayesian classifier in which allelic states across *P. falciparum* clones are estimated explicitly using a Gibbs sampler and the likelihood of recrudescence is averaged over each pairwise comparison of clones in the baseline and recurrent sample[8]. An informal consultation convened by WHO in 2021 recommended study of "the impact of Bayesian analysis on recrudescence rates in areas of high transmission, given the trend in increased recrudescence rates using this method, to determine whether this is an artefact of the method, whether there is some reason for higher failure rates in these areas (e.g., a high MOI may be more challenging for antimalarial drugs to clear), or whether there is emergence of true antimalarial resistance that needs rigorous confirmation"[9].

In this work, we use microsatellite data provided from 70 paired *P. falciparum* recurrent infections from a study conducted in Angola in 2021[13] to evaluate the CDC Bayesian classifier and explore the impact of model misspecification on recurrence classification. We construct a novel probabilistic Bayesian classifier (implemented in the R software PfRecur) for paired genotyping data, allowing recurrent infections to be mixtures of recrudescent and newly-inoculated clones[9,14]. We show that this formulation makes PfRecur robust to misspecification in the likelihood and it solves combinatorial problems for multiclonal samples, both for the pairwise comparisons between baseline and recurrent samples[15], and the computation of sample allele frequencies[16–20]. Because the posterior probability is analytically tractable, PfRecur is fit to data analytically. This precludes the need for an optimisation algorithm or numerical sampler. We compare the empirical performance of simple match-counting algorithms versus the CDC Bayesian classifier and our novel approach PfRecur. We show that in high transmission areas, where infections are often polyclonal, the CDC Bayesian classifier and match-counting approaches may yield unacceptably high false positive rates. This can result in overestimation of treatment failure rates and thus overestimation of antimalarial drug resistance, raising concerns and potentially prompting unnecessary and expensive changes in treatment policy[21,22].

## Results

### PfRecur: a probabilistic model of recrudescence
We developed a novel probabilistic model for distinguishing *P. falciparum* recrudescence versus reinfection based on observed alleles from multiple markers in paired baseline and recurrent infections; for simplicity, we refer to this model as PfRecur. Rather than treating reinfection and recrudescence as mutually exclusive categories, each recurrent infection is modelled as a mixture of newly-inoculated (from reinfection) and recrudescent parasite clones. This construction is motivated by a concern for robustness against model misspecification. Under the simplifying assumptions of marker-wise and clone-wise independence (within samples and between non-recrudescent clones across samples), we derive an analytic per-patient likelihood that averages over all allelic configurations that are compatible with the sample MOI and the set of observed genotypes in each sample. We accommodate undetected clones in baseline/recurrent samples for the patient of interest only, whereby each clone is modelled to be genotyped (i.e, detected) with probability $\omega$ at each marker; this yields a marker-wise truncated binomial model for the number of clones that have contributed to the observed set of genotypes at each marker. Site-specific allele frequencies for newly-inoculated clones are derived under a multinomial-Dirichlet model from the baseline samples, assuming exchangeability among study patients (excluding the patient of interest). These site-specific allele frequencies are also used to impute allelic states at each marker for undetected clones in the baseline sample for each patient. Allele frequencies in the recrudescent clones are based on the paired patient baseline clones following imputation. We adjust for genotyping error in baseline samples, relative to the recurrent sample of interest, through a non-parametric marker-agnostic model $\delta_\ell$ governing the probability that each allele called in a baseline sample matches an allele called in the recurrent sample. In this study, we consider a normalised geometric error model (with respect to allele repeat lengths, akin to ref. [8]) with error probability $\varepsilon$, adapted to length polymorphic markers. $\omega$, $\varepsilon$ are user-specified parameters in the model with default values of 0.9 and 0.05, respectively.

Under a Bayesian framework, with a symmetric beta binomial prior for the number of newly-inoculated versus recrudescent clones within a recurrent sample, we derive an analytical posterior distribution for the number of recrudescent clones within each recurrent sample. We consider two posterior summary metrics: a) the posterior probability of there being at least one recrudescent clone in the recurrent sample (denoted M1); and b) the expected proportion of recrudescent clones in the recurrent sample (M2). These metrics are identical for recurrent samples with a single clone (i.e., MOI = 1). An extended comparison of the PfRecur model structure against the Bayesian CDC model[8] is provided in Supplementary Note 4.

### Simulation study
We conducted a simulation study to evaluate the ability of our probabilistic classifier PfRecur to resolve mixtures of newly-inoculated and recrudescent clones. Simulated data (Supplementary Note 2) were well-specified relative to our classification model apart from three features: first, non-meiotic siblings (i.e., genetically related parasites derived from independent crosses of the same parental pair) could be present within samples (thereby violating the assumption of clone-wise independence within samples); second, the observed MOI of each sample (i.e., the maximum observed cardinality across simulated markers) could be lower than the true MOI (because of undetected clones or allelic overlap between clones); and third, the detection of clones in baseline samples (used to derive allele frequencies for newly-inoculated clones) was incomplete. We considered an idealised setting, where both genotyping error probabilities $\varepsilon$ and the per-clone marker-wise detection probability $\omega$ under which the data were generated were known.

Using metric M2, our model PfRecur was able to recover the proportion of recrudescent clones within simulated recurrences, albeit with decreasing precision for recurrences with high MOIs (Fig. 1B). Metric M1 (i.e., the posterior probability of at least one recrudescent clone) had very high sensitivity in detecting recurrences with ≥1 recrudescent clone,

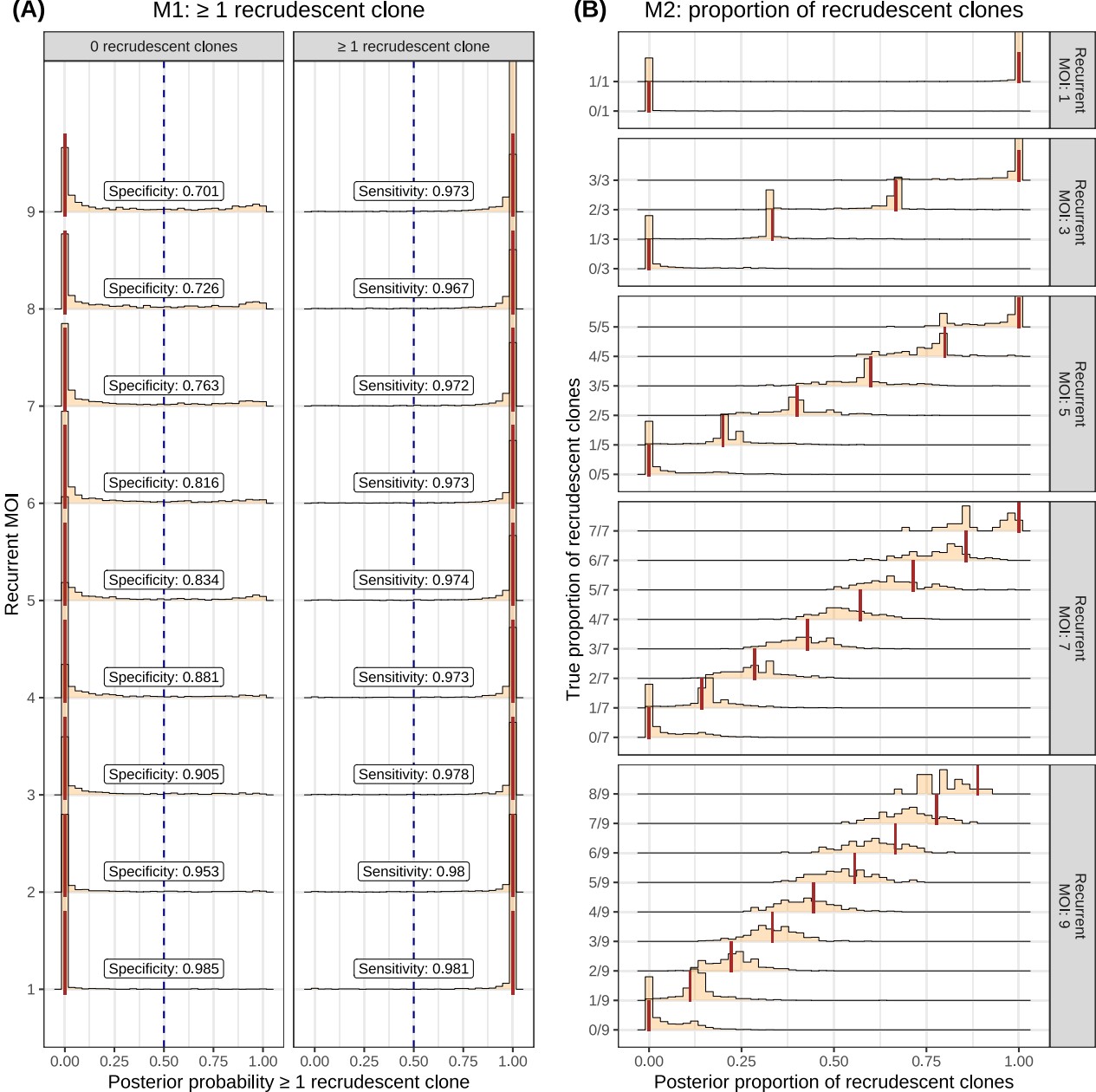

**Fig. 1 | Application of PfRecur to simulated mixtures of newly-inoculated and recrudescent *P. falciparum* clones. A** shows the posterior probability of at least one recrudescent clone (M1) (annotated with classification sensitivity and specificity using a threshold of 0.5 to call recrudescence), while **B** shows the posterior proportion of recrudescent clones (M2), aggregated over 29077 simulated recurrences. Truth values are shown with vertical lines.

but was liable to call false positive recrudescences using a threshold of 0.5 (dashed blue line) at MOIs above 6 (Fig. 1A). For recurrences with an elevated MOI of *m*, classification using metric M2 with a threshold of 1/*m* has higher specificity than using metric M1 with a threshold of 0.5.

### Therapeutic efficacy study in Angola[13]

We used open access data from a therapeutic efficacy study conducted in Angola in 2021[13] to compare PfRecur with match-counting algorithms and the CDC model. The Angolan study evaluated four anti-malarial drugs: artemether-lumefantrine (AL), artesunate-amodiaquine (ASAQ), dihydroartemisinin-piperaquine (DP), and artesunate-pyronaridine (ASPY), across 3 study sites in 622 patients with uncomplicated malaria[13]. Recurrent parasitaemia was detected in 71 patients between 7 and 42 days of follow-up (Table 1). Paired baseline and recurrent *P. falciparum* samples (70 pairs) were genotyped at 7

neutral microsatellite markers (*M2490, M313, M383, PfPK2, POLYA, TA1, TA109*). An additional 14 unpaired baseline samples per study site were also genotyped to increase the precision of the estimation of population allele frequencies, stratified by study site. Transmission intensities varied across the three sites, with Zaire and Lunda Sul classified as high and moderate intensity transmission, respectively, and Benguela as low transmission intensity. In Zaire and Lunda Sul combined within 28 days of follow-up there were 36/198 (18%) recurrent infections in the AL treated patients and 19/188 (10%) recurrent infections in the artesunate-amodiaquine (ASAQ) treated patients. In Benguela, dihydroartemisinin-piperaquine (DP) and artesunate-pyronaridine (ASPY) were compared, and 14/100 and 6/104 recurrent infections were observed within 42 days of follow-up, respectively. Under the CDC model, this translated into estimated day-28 PCR-corrected efficacies of 88% (95% CI 82–95) and 94.4% (95% CI 90–99) for AL; 91%

**Table 1 | Summary of a therapeutic efficacy study conducted in Angola in 2021 by[13] (adapted from Tables 1 and 3 of[13])**

| | Benguela | | Zaire | | Lunda Sul | |
|---|---|---|---|---|---|---|
| | ASPY | DP | AL | ASAQ | AL | ASAQ |
| Study period | Mar–May 2021 | May–Jul 2021 | Feb–Mar 2021 | Mar–Jun 2021 | Feb–Apr 2021 | Mar–Jun 2021 |
| No. patients enrolled | 104 | 105 | 104 | 105 | 104 | 100 |
| No. patients completed follow-up | 100 | 104 | 98 | 97 | 100 | 91 |
| Median age, years (range) | 6.9 (0.5–12) | 7 (0.6–12) | 2.5 (0.5–5) | 2.5 (0.7–5) | 2.6 (0.5–5) | 2.7 (0.6–5) |
| % patients female | 49% | 47% | 45% | 56% | 49% | 46% |
| No. early treatment failures (day <7) | 0 | 0 | 1 | 3 | 0 | 0 |
| No. late treatment failures (day ≥7) | 14[a] | 6 | 22 | 16 | 13 | 0 |
| (with baseline MOI > 1[b]) | (3[a]) | (4) | (12) | (7) | (10) | (0) |

*ASPY* artesunate-pyronaridine, *DP* dihydroartemisinin-piperaquine, *AL* arthemether-lumenfantrine, *ASAQ* artesunate-amodiaquine.
[a]Genotypes are missing for one recurrent sample from the ASPY arm in Benguela.
[b]Based on the maximum observed cardinality across 7 neutral microsatellite markers for the day 0 sample.

(95%CI 85–97) and 100% for ASAQ; and day-42 PCR-corrected efficacies of 99.6% (95% CI 99–100) for DP and 98.3% (95% CI 96-100) for ASPY. As AL is the most frequently used ACT in the world[23], the low efficacy estimate of AL in the Zaire site was of particular concern.

## Artefacts which occur at low malaria parasite densities

Visual inspection of the genotypic data from[13] showed that some of the PCR molecular markers were much more likely to be polyallelic than others. Exploratory analysis demonstrated that the parasite density ($\log_{10}$ parasitaemia) was strongly predictive of the observed MOI (maximum cardinality across the 7 markers; Fig. 2A). Low parasitaemia (<1000 per µL) was associated with MOIs of 2 or more. When assessing individual markers, it was apparent that the microsatellite *TA109* is problematic, particularly at low parasitaemias (Fig. 2B), yielding elevated apparent MOIs for recurrences with parasitaemia below 1000/µL. This suggests that *TA109* multiplicity (and thus MOIs based on this marker) in low parasite density samples may be artefactual. This likely results from methodological or laboratory artefacts (e.g., non-specific peaks). We therefore conducted all analyses with and without *TA109* to test for robustness of the methodology relative to inclusion of an unreliable polymorphic marker.

## Estimation of *P. falciparum* recrudescence for samples with baseline MOI > 1

We estimated the probability of recrudescence for each patient with a recurrent infection in the Angola study[13], using the CDC model and PfRecur. We used metric M1 (probability of at least one recrudescent clone) for conceptual consistency with the CDC model[8]. For paired acute and recurrent infections with a baseline MOI of 1, the probabilistic models do not differ substantially. Both models correct for chance allelic matches, which is an advantage over a simple match-counting algorithm. However, when the MOI of the baseline infection exceeds 1, issues of multiple comparisons become important. Figure 3 shows the model estimates for the 36 paired infections in the Angolan study with baseline MOI > 1, ordered by the posterior probability of recrudescence under the CDC model. For a third (12/36) of the recurrent infections, there was a non-negligible difference in the model estimates, with the CDC model generally estimating higher recrudescence probabilities. Discrepancies were largely apparent for recurrences with intermediate posterior probabilities under the CDC model, and many of these differences persisted after exclusion of the problematic microsatellite marker *TA109*. Visual inspection of the samples where the models differ substantially suggest that the CDC model is over-calling recrudescence (Supplementary Fig. 5).

## Estimation of false positive *P. falciparum* recrudescence rates

We used a permutation method to estimate false positive rates for recrudescence classification. We generated 500 artificial 'not-

recrudescence' datasets by shuffling the participant identifiers of the baseline samples in ref. 13, stratified by study site and thus preserving population structure. Figure 4 shows the estimated false positive rates, calculated by averaging metric M1 of PfRecur and the posterior probability of recrudescence under the CDC model across recurrent samples. Across the permuted datasets, the CDC model had median false discovery rates of 8.7% (95% confidence interval [CI] 1.2–19.1) in Benguela, 6.5% (95% CI 0.7–18.6) in Lunda Sul and 5.0% (95% CI 1.3–10.2) in Zaire. This was driven largely by permuted pairs with intermediate posterior probabilities of recrudescence (Supplementary Fig. 4). In comparison, PfRecur had median false discovery rates which were less than half those of the CDC model: 4.1% (95% CI 0.3–13.1) in Benguela, 1.2% (95% CI 0.1–9.1) in Lunda Sul and 1.6% (95% CI 0.1–6.1) in Zaire. The false discovery rates using PfRecur remained lower than with the CDC model even as the per-clone marker-wise probability of detection was relaxed from the default value $\omega = 0.9$ to $\omega = 0.75$ (assuming more clones evade detection tends to augment the posterior probability of recrudescence) (Supplementary Fig. 3). The ≥4/7 match-counting rule recommended by refs. 13,24,25 yielded higher false discovery rates than PfRecur in Benguela and Zaire.

We note that the false discovery rates are dependent in part on marker diversity: in settings with limited diversity, we would expect PfRecur to return the prior distribution over newly-inoculated versus recrudescent clones, and the CDC model to return a 0.5 posterior probability of recrudescence, yielding an elevated apparent false positive recrudescence rate using this permutation method.

## Discussion

Accurate characterisation of ACT failure rates in uncomplicated malaria is essential to guide policies and practices, especially now in Africa where artemisinin resistant *P. falciparum* is spreading and consequently ACTs are under increasing threat[26]. Genotyping of polymorphic alleles has allowed the clinical evaluation of antimalarial therapeutic efficacy in malaria endemic areas where study participants may develop new infections during follow-up. But in high transmission settings differentiating between reinfection and recrudescence remains a difficult problem. Many malaria infections are polyclonal, posing combinatorial problems since allele counts are not directly observable. The different *P. falciparum* PCR genotyping methods also have different sensitivities[10]. Several different statistical approaches have been proposed. Match-counting methods – which do not adjust for multiple comparisons across polyclonal samples, relative allele frequencies[8,11] or the imperfect detectability of parasite clones[27,28] – have limitations which are well-established in the literature[24,28,29]. In addition to requiring good laboratory techniques for accurate genotype calling, a robust statistical methodology is needed to assess probabilistically whether the recurrent infection is compatible with recrudescence (treatment failure). The statistical model cannot solve all technical and biological complexities,

**(A)**

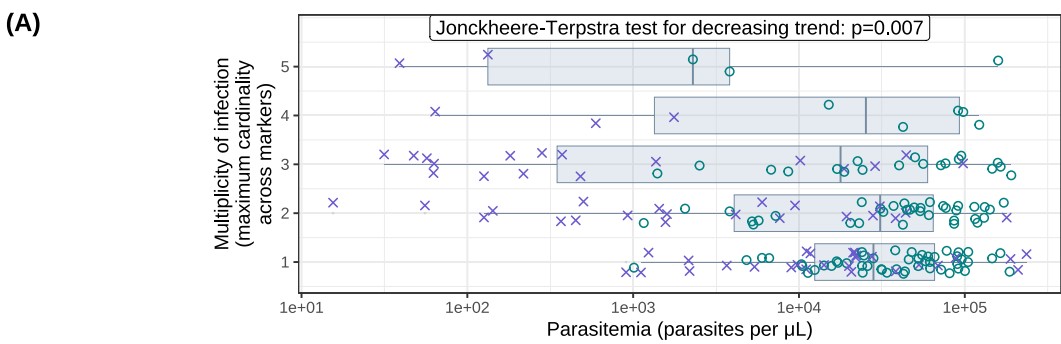

**(B)**

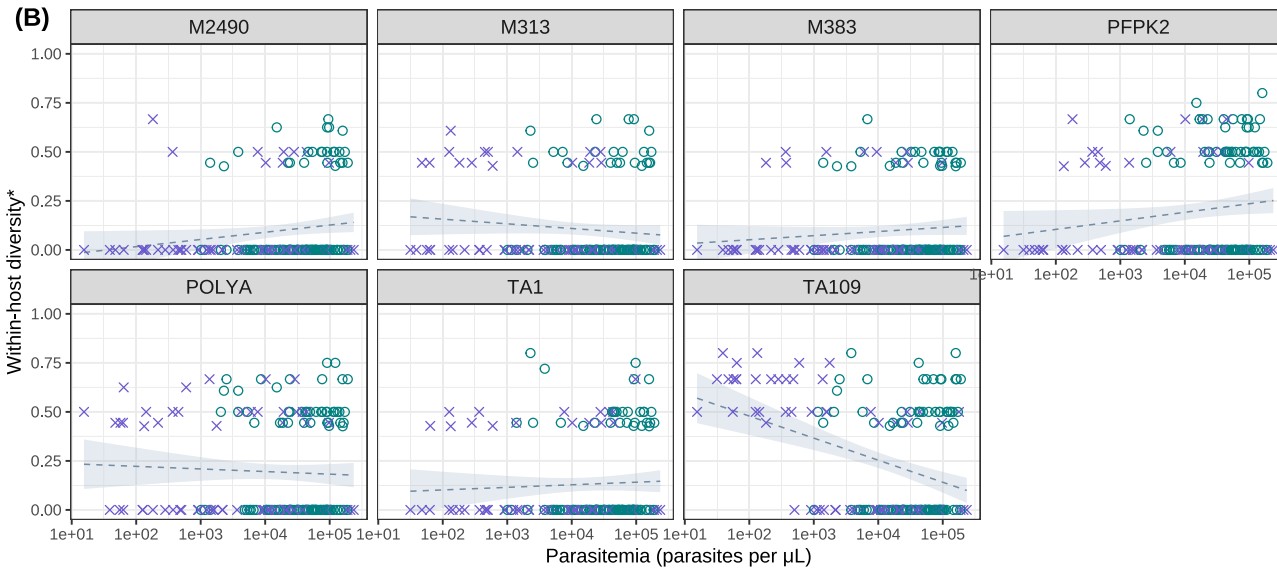

**Fig. 2 | Marker cardinality and parasitemia in ref. [13]. A** Parasite density stratified by the multiplicity of infection (i.e., the maximum cardinality across 7 neutral microsatellite markers) for 182 blood samples with microscopy-detectable asexual *P. falciparum* parasitaemia genotyped by[13], comprising 70 of 71 detected recurrent infections (characterised by the presence of detectable parasitaemia on or after day 7 of follow-up); the 70 corresponding baseline (day 0) infections; and 42 additional baseline infections. Box limits indicate upper and lower quartiles; centre lines correspond to medians; whiskers are truncated at the range, or 1.5 times the width of the interquartile range. The Jonckheere-Terpstra test[34] for a decreasing trend in

MOI as a function of $\log_{10}$ parasitemia yields a J statistic of 7278 with standardised Z-score -2.4748, and *p*-value 0.006665 derived under a normal approximation[35]. **B** The within-host diversity of each microsatellite marker shown as a function of parasite density. We define within-host diversity as the expected value of 1-Nei's gene identity metric, assuming *M* equifrequent clones (where *M* is the maximum cardinality across all loci) exhibit *A* distinct alleles (where *A* is the locus cardinality). Dashed lines show the line of best fit, while shaded bands show 95% confidence intervals based on linear regression of the marker-wise within-host diversity metric against $\log_{10}$ parasitemia.

but it should provide a principled framework for handling paired genotypic data from polyclonal infections. Complexities which would confound interpretation of outcomes, such as the presence of an undetected asynchronous clone which was sequestered at the time of treatment and later caused a recrudescent infection[12], are out of scope for a general purpose statistical model.

We show that when *P. falciparum* infections are polyclonal, match-counting methods and the methodology proposed by the CDC[8] may result in unacceptably high false positive recrudescence rate estimates. The PfRecur classification model has higher specificity whilst retaining good accuracy at identifying recrudescence. When applied to microsatellite data from a recent therapeutic efficacy assessment in Angola[13], the PfRecur classification model outputs different estimates for a third (12/36) of the recurrent infections with baseline MOI > 1[8]. However, this did not have a large effect on efficacy estimates across study arms (Supplementary Table 3). The systematic overestimation of recrudescence rates is likely to be greater in settings with greater parasite diversity, higher polyclonality, and more frequent reinfection.

The Bayesian model proposed by the CDC[8] has three major issues. First, a per-clone unobserved allele penalty − a multiplicative factor

applied to the likelihood when the imputed allele for a clone lies outside the observed set of alleles for a given sample − is estimated simultaneously during classification (a relatively lax penalty in the range 0.25 to 0.45 was estimated for each study site in ref. [13]). We suggest that this construction increases the estimated probability of recrudescence. This is because although the unobserved alleles have little effect on the probability of reinfection (they only affect allele frequency estimates), they do tend to increase the chance of allelic matches between the paired samples and this therefore increases the estimated probability of recrudescence. This bias is particularly pronounced for samples with no genotyping data at a subset of markers (which is more likely at low parasite densities), and samples with unbalanced cardinality across loci. Markers with missing genotypes are included in the estimation procedure, and the per-clone unobserved allele penalty is applied irrespective of whether or not imputed alleles match those in the paired sample. Samples with unbalanced cardinality across loci result in a greater chance of introducing unobserved alleles, a feature that may be problematic when MOIs are inflated as a result of artificially high multiplicities at one marker. This occurred with the *TA109* microsatellite in the dataset analysed here. Second, estimation

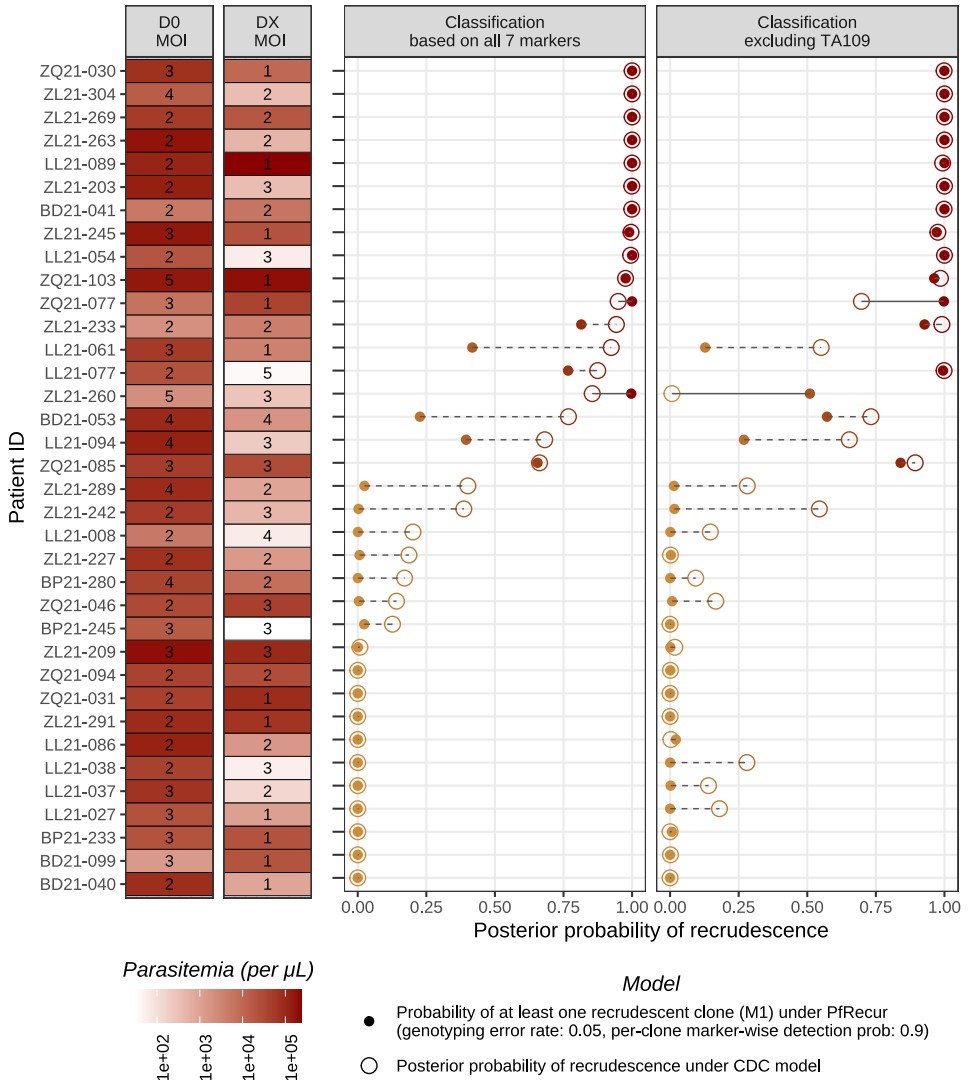

**Fig. 3 | Summary of posterior estimates for recrudescences in *P. falciparum* recurrences reported by[13] with baseline MOI > 1.** We compare the probability of at least one recrudescent clone under PfRecur (with genotyping error probability $\varepsilon = 0.05$ under the normalised geometric model (7) and marker-wise per-clone detection probability $\omega = 0.9$, applied to clones in the pair of baseline and recurrent samples for the participant of interest) against the posterior probability of recrudescence based on the CDC model[8]. Under PfRecur, allele frequencies for newly-inoculated clones within each recurrence have been derived from available genotypes for baseline samples from the same study site, excluding the patient of interest. There were 32 unpaired baseline samples available for Benguela (B); 26 unpaired baseline samples from Lunda Sul (L) and 51 unpaired baseline samples from Zaire (Z).

of the genotyping error appears to be unstable. In the dataset analysed, site specific estimates of genotyping error varied from 0.025 in Lunda Sul to 0.072 in Benguela (i.e., a three fold difference) although all genotyping was presumably done by the same central laboratory. The difference in estimates suggests there may be identifiability issues in the model. As demonstrated by the marker *TA109*, a major determinant of genotyping error is likely to be the parasite density in the sample tested. Third, the reliability of the output model probabilities depends on convergence of the algorithm. To the best of our knowledge, there has been no in depth study of convergence of this algorithm, and the default parameters suggested (1000 iterations[30]) appear insufficient. In contrast, PfRecur averages over compatible allelic configurations within each sample – accounting for the imperfect detection of clones in the paired baseline and recurrent samples under a marker-wise truncated binomial model governed by the user-specified per-clone probability of detection $\omega$, and user-specified marker-wise non-parametric genotyping error matrices – to evaluate directly an analytically-tractable (discrete) posterior distribution for

the number of newly-inoculated vs recrudescent clones within each recurrent sample, allowing for classification of recurrent samples with multiplicity of infection up to 9 in the order of seconds.

A limitation of PfRecur in low transmission settings is the assumed lack of relatedness structure within or between samples (the CDC model[8] makes the same assumption). The classifier of ref. 31 for *P. vivax* recurrence explicitly accommodates sibling relationships within samples (and between samples, but only in support of relapse identification), and in addition, includes an optional fudge-factor for low-level background relatedness; but the transitive property of relatedness introduces substantial complexity. A recent classification model developed by ref. 32 accommodates estimates of background relatedness in the classification of *P. falciparum* recurrence. However, we show in ref. 33 that sample allele frequencies, which are used in many models of malaria parasite genetic data, can be thought to implicitly encode average relatedness marker-wise.

In conclusion, we present an analytically-tractable probabilistic classifier PfRecur for estimating recrudescence based on multi-allelic

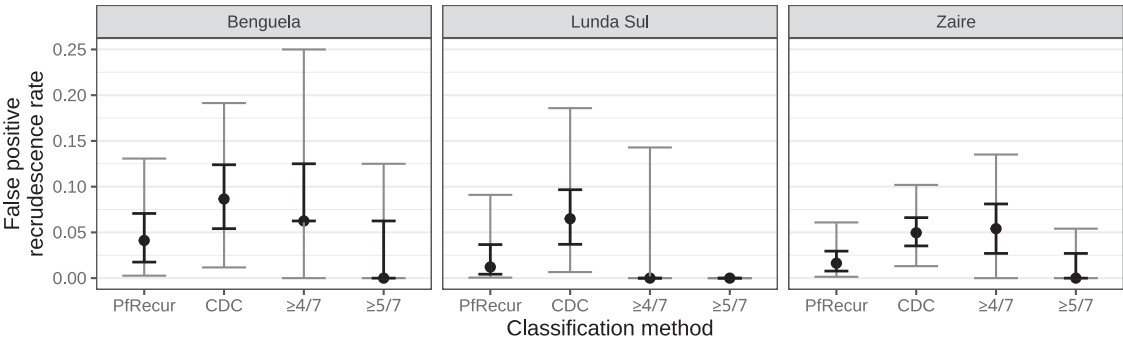

**Fig. 4 | False positive recrudescence rates based on artificial permuted 'not-recrudescence' data generated from[13].** For PfRecur we average metric M1; for the CDC model[8], we average the posterior probability of recrudescence. For match-counting, we treat the presence of one or more shared alleles at 4 or more (≥4/7)[13,24,25], or 5 or more (≥5/7), markers as evidence of recrudescence (match-counting is restricted to permuted pairs with ≥1 allele call at each of the 7 markers). Points indicate medians, bold error bars indicate the interquartile range and grey error bars indicate 95% confidence intervals over 500 permuted artificial 'not-recrudescence' datasets, each comprising 19 recurrent and 33 baseline samples from Benguela; 13 recurrent and 27 baseline samples from Lunda Sul and 38 recurrent and 52 baseline samples from Zaire[13]. For match-counting, there are 16 recurrent samples from Benguela, 7 recurrent samples from Lunda Sul and 37 recurrent samples from Zaire with at least one allele call at each of the 7 markers.

calls in recurrent falciparum malaria. In high transmission settings it may be more accurate than current methods of analysis and may therefore be less likely to overestimate recrudescence rates.

## Methods
### Data from[13]
We re-analysed data from a six-arm therapeutic efficacy study conducted between February and July in 2021 across 3 provinces in Angola[13]. For completeness, we provide a summary of the study design and genotyping approach of ref. 13 here. Outpatients presenting to urban clinics situated in provincial capitals were screened for inclusion in the study. Enrolment criteria included uncomplicated *P. falciparum* monoinfection and either a history of fever or an axillary temperature reading ≥37.5 °C. To avoid confounding by different transmission intensities, some inclusion criteria varied by province. In Benguela (low to moderate transmission), children between 6 and 143 months of age with 1000−100,000 parasites/$\mu$L at baseline were enrolled. In Lunda Sul (moderate to high transmission) and Zaire (moderate to very high transmission), children were enrolled with a narrower age range of 6 to 59 months, and higher baseline parasite densities of 2000−200,000 parasites/$\mu$L. Parasitaemia was quantified using microscopy. 622 patients were enrolled across the study arms.

Study participants were treated with 3-day regimens of either artemether-lumefantrine or artesunate-amodiaquine (Lunda Sul and Zaire), and dihydroartemsinin-piperaquine or artesunate-pyronaridine (Benguela). Dosing was determined by weight bands in accordance with manufacturer's guidelines. Antimalarial treatment was largely supervised. Follow-up, entailing clinical examination and slide microscopy, occurred on days 1 (clinical examination only), 2, 3, 7, 14, 21 and 28 in addition to days 35 and 42 for patients treated with dihydroartemsinin-piperaquine and artesunate-pyronaridine, with the convention that enrolment (baseline) was designated day 0. Recurrent infections, characterised by microscopy-detected asexual *P. falciparum* parasitaemia occurring between day 7 and the end of follow-up, were identified in 71 patients.

Genotyping was performed for 70 pairs of baseline and recurrent samples, and an additional 42 baseline samples. DNA was extracted from dried blood spots and *P. falciparum* diagnosis was confirmed by PCR. For a panel of 7 neutral microsatellite markers (*M2490, M313, M383, PfPK2, POLYA, TA1, TA109*), fragment lengths were then assessed using capillary electrophoresis. In the original study[13], classification of reinfection versus recrudescence was performed using the Bayesian CDC model[8], and a simple match-counting algorithm (at least 4/7 matches[13,24,25], where a per-marker observation is a match if some

alleles are the same at enrolment and recurrence), stratified by study site. Here, we also perform classification using our novel probabilistic approach, PfRecur and a 5/7 matching algorithm.

### Molecular marker cardinality and parasitaemia in[13]
We explored the relationship between MOI (defined as the maximum cardinality across the genotyped panel of 7 neutral microsatellites) and $\log_{10}$ parasitemia for 182 samples with genotyping data available, using the Jonckheere-Terpstra test[34], with *p*-values derived under a normal approximation using the R function `PMCMRplus::jonckheereTest` (V1.9.6)[35]. We also examined the within-host diversity of each marker, as a function of $\log_{10}$ parasite density, performing linear regression using the R functions `ggplot2::geom_smooth` (V3.5.1)[36] and `stats::lm` (V4.2.1)[37]. We defined the within-host diversity of each marker for each sample as the complement of Nei's gene identity metric[38], taking a uniform distribution over compatible allelic configurations at that locus (i.e., all possible ways of allocating alleles observed at that locus to the number of clones given by the MOI). Nei's gene identity metric is formulated as the sum of squares of allele frequencies; the complement can be interpreted as the probability of differing alleles when a pair of clones is sampled (with replacement) from the sample (see Supplementary Note 3.1).

### PfRecur
PfRecur classifies a recurrent infection as either a recrudescence or a reinfection based on either the posterior probability of there being at least one recrudescent clone (M1); or the posterior expected proportion of recrudescent clones (M2) (Fig. 5). A complete description of the framework is provided in Supplementary Note 1; below, we provide a brief outline, using the terms marker and locus interchangeably. Classification of recurrent infections is done for each pair in a group of patients, where the grouping is determined by the user (usually by study or study site, depending on the context). The model is not fully Bayesian in that there is no way of specifying multilevel groupings.

**Overview of the statistical model.** PfRecur classifies infections based on an analytically tractable Bayesian model. Under the model, each sample is treated as a set of genetically-distinct clones; the cardinality of this set is referred to hereafter as the multiplicity of infection (MOI). For each sample, we observe a set of alleles $G_\ell$ at loci $\ell \in \{1, ..., L\}$, with locus-wise cardinality $M_\ell = |G_\ell|$. We take the MOI to be the maximum cardinality over all loci: $M = \max_{1 \leq \ell \leq L} M_\ell$.

The indices *r*, *b* and *i* are used to distinguish samples that are handled differently under the model. Index *r* corresponds to the

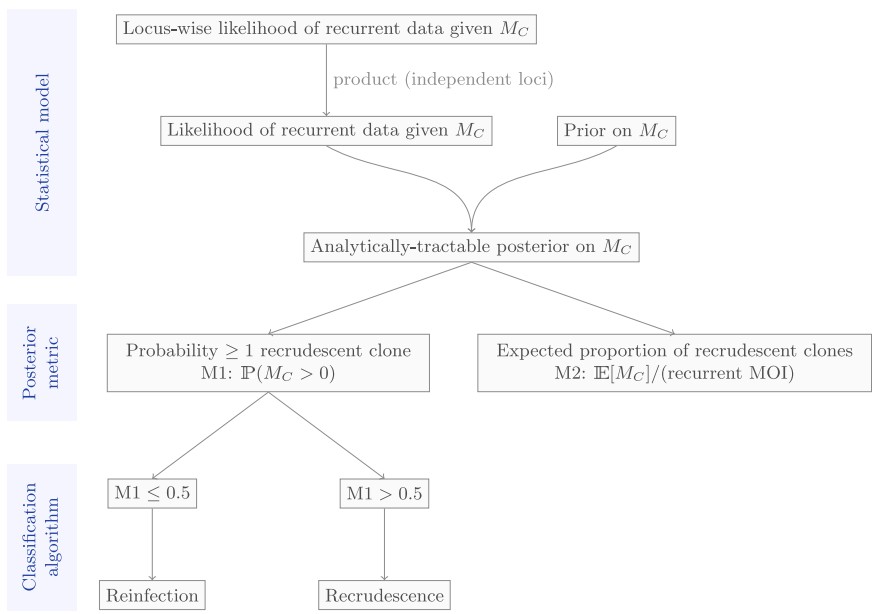

**Fig. 5 | Overview of the PfRecur model framework.** PfRecur is designed to classify a recurrence as either a recrudescence or a reinfection using a posterior summary of $M_C$, the number of recrudescent clones in the $r$th sample.

recurrent parasite sample for which classification is being performed; index $b$ corresponds to the paired baseline sample (i.e., it is from the same study participant as sample $r$). Index $i = 1, ..., n$ iterates over $n$ baseline samples that are not paired with sample $r$: they are from different study participants.

The recurrent sample $r$ (of MOI $M^{(r)}$) is modelled as a mixture of $M_C$ recrudescent clones and $M_I = M^{(r)} - M_C$ newly-inoculated clones. For each recurrent sample $r$, the target of inference under the PfRecur statistical model is the posterior distribution over $M_C$, with state space $\{0, ..., \min(M^{(b)}, M^{(r)})\}$:

$$\mathbb{P}\left(M_C = m \mid G^{(r)}\right) = \frac{\mathbb{P}(M_C = m) \prod_{\ell=1}^{L} \mathbb{P}\left(G_\ell^{(r)} \mid M_C = m, M_I = M^{(r)} - m\right)}{\sum_{m=0}^{\min(M^{(b)}, M^{(r)})} \mathbb{P}(M_C = m) \prod_{\ell=1}^{L} \mathbb{P}\left(G_\ell^{(r)} \mid M_C = m, M_I = M^{(r)} - m\right)}.$$
(1)

The model is predicated on the following set of assumptions:

- *Unlinked loci*: valid if neutral loci lie on different chromosomes (because chromosomes assort independently in meiosis) or inter-locus distances are large (because then loci are more likely to be separated by one or more recombination break points in meiosis).
- *Clones within samples are independent as are non-recrudescent clones between samples*: violated by the presence of sibling parasites within samples[31]; additionally violated if the average population-level relatedness is high (although sample allele frequencies for the baseline samples 1, ..., $n$ partially encode average relatedness[33]) and/or if the population is structured (e.g., by geographic barriers in study site, or by household effects among study participants).
- *Uniform distribution of allelic states for each sample* (i.e., any configuration of alleles compatible with the number of successfully genotyped clones and the set of genotypes observed at a given locus is modelled to be equally likely): enforced in the absence of quantitative data on relative allelic abundance (i.e., assuming bulk genotypic data with no quantitative information on the intra-sample amount of each allele), given measures of allelic abundance are not readily available for length polymorphic markers that are genotyped using electrophoresis and currently used to perform molecular correction; however, we note that amplicon sequencing, which is recognised by the WHO as a potential future standard for molecular correction, generates read count data on multi-allelic

markers (microhaplotypes), and read count data can be used to estimate the relative abundance of within-sample alleles.

- *Reinfection and recrudescence are not mutually-exclusive*: the recurrent sample $r$ comprises a mixture of recrudescent clones drawn from the paired baseline sample $b$, and newly-inoculated clones drawn from the contemporaneous population at large, which is approximated by the remaining baseline samples 1, ..., $n$.
- *Non-parametric genotyping error*: we accommodate non-parametric genotyping error in baseline samples relative to the recurrent sample, specified by the probability that each allele called in a baseline sample matches an allele called in the recurrent sample $r$.
- *No undetected clones in the baseline samples* 1, ..., $n$: we assume that the consequences of ungenotyped clones (if any) 'average out' over samples 1, ..., $n$ from which baseline population allele frequencies are derived.
- *Ungenotyped clones in the paired baseline sample $b$*: the allelic states of undetected clones in the paired baseline sample are imputed using allele frequencies derived over the unpaired baseline samples 1, ..., $n$, with a marker-wise truncated binomial model for the number of clones genotyped per locus.
- *Ungenotyped clones in the recurrent sample $r$*: observed alleles in the recurrent sample are allocated over successfully genotyped clones only, with a marker-wise truncated multinomial model for the number of clones genotyped per locus.

**Model structure.** Figure 6 depicts the model structure. The likelihood of the locus-wise recurrent data, $G_\ell^{(r)}$, is conditional on the random variables $M_C$ and $M_I = M^{(r)} - M_C$. It is also conditional on the locus-wise baseline data, $G_\ell^{(b)}$ and $\{G_\ell^{(i)}\}_{i=1..n}^{i \neq b}$, and on the recurrent and baseline MOIs, which are derived from the complete recurrent and baseline data. Although these variables are data derived, they are not treated as random variables under the model. The likelihood also features other user-defined inputs that are not treated as random variables. These include a genotyping detection rate, $\omega$, and a locus-wise genotyping error matrix, $\delta_\ell$. The prior on $M_C$ is conditional on $M^{(r)}$ and on a prior parameter $\beta$. There is no prior on $M_I$ in Equation (1) since $M_I$ is deterministic given $M^{(r)}$ and a realisation of $M_C$. Thanks to an analytically tractable likelihood (next section), Equation (1) is analytically tractable. As such, inference under the PfRecur framework does not require a numerical sampler or an optimisation algorithm.

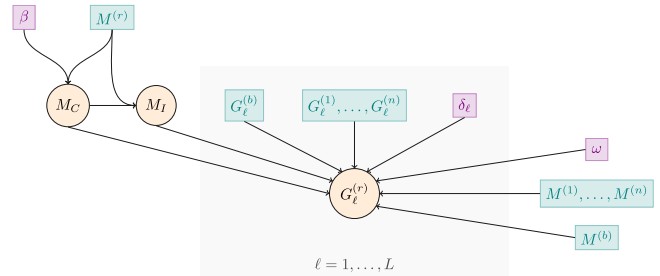

**Fig. 6 | Graphical representation of the Bayesian statistical model within the PfRecur framework.** Random variables are circled. Data driven quantities are shown in teal; user-specified values are shown in violet.

**Model likelihood.** We derive multiple expressions in Supplementary Note 1 (see Supplementary Equations (15), (17), (18) and (19)), which together can be used to evaluate the likelihood:

$$\mathbb{P}\left(G_\ell^{(r)} \mid M_c = m, M_I = M^{(r)} - m\right). \tag{2}$$

The various steps used to conceptualize the likelihood are sketched out below. Throughout, newly-inoculated clones are drawn from population $I$ (the contemporaneous population at large), which is approximated under the model by the baseline samples $i = 1, \ldots, n$. The recrudescent clones are drawn from population $C$, comprised of clones in the paired baseline sample $b$. In an intermediate step, the probability of $G_\ell^{(r)}$ is modelled conditional on allele frequencies, whereby the probability that a clone drawn from population $S \in \{I, C\}$ harbours allele $\alpha$ at locus $\ell$ is equated to the population allele frequency $\theta_\ell^{(S)}(\alpha)$. In later steps, allele counts supersede allele frequencies.

**Modelling the number of clones genotyped per locus.** In the allelic data of a given sample, some loci have cardinalities lower than the MOI. The lower cardinality could occur because multiple clones within the sample share identical alleles at this locus, because the MOI is spuriously elevated by genotyping errors, or because some clones are not genotyped at this locus. We model the number of genotyped clones per locus in the recurrent sample $r$ at loci with cardinalities strictly less than the observed MOI (i.e., $|G_\ell^{(r)}| < M^{(r)}$), thereby allowing some clones in the recurrent sample to evade detection at some loci (multiple clones sharing identical alleles is also addressed – see next section). A spuriously elevated MOI is not accounted for.

We construct a model of the number of clones genotyped per locus in the recurrent sample $r$, i.e., the number of clones that contribute to the observation $G_\ell^{(r)}$, as follows. Each clone in sample $r$, irrespective of whether it is drawn from population $I$ or $C$, is detected with probability $\omega$ at locus $\ell$. Then, given $M_C = m$ and $M_I = M^{(r)} - m$, the number of detected clones $Q_\ell^{(r, S)}$ derived from populations $S \in \{I, C\}$ at locus $\ell$ follow the truncated multinomial distribution

$$\mathbb{P}\left(Q_\ell^{(r, C)} = q_C, Q_\ell^{(r, I)} = q_I, \mid M_C = m, M_I = M^{(r)} - m\right)$$

$$= \frac{\binom{m}{q_C}\binom{M^{(r)} - m}{q_I}\omega^{q_I + q_C}(1 - \omega)^{M^{(r)} - q_I - q_C}}{\sum_{j = M_\ell^{(r)}}^{M^{(r)}} \binom{M^{(r)}}{j}\omega^j(1 - \omega)^{M^{(r)} - j}} \text{ for } q_I + q_C \geq M_\ell^{(r)}, q_C \leq m, q_I \leq M^{(r)} - m. \tag{3}$$

**Allocating observed alleles to genotyped clones.** We average analytically over compatible allelic configurations in sample $r$ using the inclusion-exclusion principle. Suppose there are $Q_\ell^{(r, S)} = q_S$ genotyped clones drawn from populations $S \in \{I, C\}$ at locus $\ell$ in sample $r$. Then, the probability of observing the set of alleles $G_\ell^{(r)}$ at locus $\ell$ takes the

form

$$\mathbb{P}\left(G_\ell^{(r)} \mid \boldsymbol{\theta}_\ell^{(C)}, \boldsymbol{\theta}_\ell^{(I)}, Q_\ell^{(r, C)} = q_C, Q_\ell^{(r, I)} = q_I\right)$$

$$= \sum_{A \in \mathcal{P}(G_\ell^{(r)})} (-1)^{|A|} \left(\sum_{\alpha \in G_\ell^{(r)} \setminus A} \theta_\ell^{(C)}(\alpha)\right)^{q_C} \left(\sum_{\alpha \in G_\ell^{(r)} \setminus A} \theta_\ell^{(I)}(\alpha)\right)^{q_I} \tag{4}$$

where $\mathcal{P}(G_\ell^{(r)})$ denotes the power set of $G_\ell^{(r)}$; that is, the set of all subsets of $G_\ell^{(r)}$ encompassing the empty set. Convolving Equation (4) over the locus-wise truncated multinomial model of genotyped clones (3) yields the probability of the observation $G_\ell^{(r)}$ at locus $\ell$ for the recurrent sample $r$, formulated with respect to the population allele frequencies $\boldsymbol{\theta}_\ell^{(S)}$, $S \in \{C, I\}$.

**Modelling allele frequencies.** The derivation of population allele frequencies under our model is two-fold. Denote by $H_\ell$ the set of possible alleles at locus $\ell$ (equipped with an arbitrary ordering) and by $\boldsymbol{\theta}_\ell^{(S)}$ the vector of population allele frequencies over $H_\ell$. We begin by deriving allele frequencies $\boldsymbol{\theta}_\ell^{(S)}$ conditional on the vector of per-sample allele counts $\mathbf{C}_\ell$ for each baseline sample $b$ or $1, \ldots, n$ over $H_\ell$; and later address the distribution of per-sample allele counts $\mathbf{C}_\ell$, which are not directly observed using bulk genotypic data.

For population $I$, we derive allele frequencies under a Bayesian multinomial-Dirichlet model of baseline samples $i = 1, \ldots, n$ (which excludes the paired baseline sample $b$). Allele frequencies are formulated over clones in the baseline samples $i = 1, \ldots, n$, whereby each sample is effectively weighted by its MOI (i.e., high MOI samples are more informative). Under the assumption of clone-wise independence (both within and across samples), we obtain the multinomial likelihood

$$\sum_{i=1}^n \mathbf{C}_\ell^{(i)} \mid \boldsymbol{\theta}_\ell^{(I)} \sim \text{Multinomial}\left(\sum_{i=1}^n M^{(i)}, \boldsymbol{\theta}_\ell^{(I)}\right).$$

Taking a uniform prior for $\boldsymbol{\theta}_\ell^{(I)}$ over the $|H_\ell| - 1$ simplex yields the posterior

$$\boldsymbol{\theta}_\ell^{(I)} \mid \sum_{i=1}^n \mathbf{C}_\ell^{(i)} \sim \text{Dirichlet}\left(\sum_{i=1}^n \mathbf{C}_\ell^{(i)} + \mathbf{1}\right). \tag{5}$$

For population $C$, allele frequencies are largely informed by the paired baseline sample $b$, but not exclusively because the alleles of ungenotyped clones in sample $b$ are imputed using population $I$ allele frequencies, which are based on samples $i = 1, \ldots, n$ (Equation (5)). This imputation explains the omission of sample $b$ in the formulation of allele frequencies for population $I$. Akin to the model of genotyped clones for sample $r$ (Equation (3)), the number of genotyped clones $Q_\ell^{(b)}$ in sample $b$ that contribute to the observation $G_\ell^{(b)}$ is modelled to follow a truncated binomial distribution, with support $|G_\ell^{(b)}|, \ldots, M^{(b)}$ and success probability $\omega$. Given $Q_\ell^{(b)} = q$, we denote by $\mathbf{C}_\ell^{(b, q)}$ the vector of allele counts over the $q$ clones in sample $b$ that are genotyped at locus $\ell$. We model $\boldsymbol{\theta}_\ell^{(C)}$ as a deterministic function of $Q_\ell^{(b)} = q$, the per-sample allele counts $\mathbf{C}_\ell^{(b, q)}$ of the genotyped clones in sample $b$, and $\boldsymbol{\theta}_\ell^{(I)}$,

$$\boldsymbol{\theta}_\ell^{(C)} = \frac{\mathbf{C}_\ell^{(b, q)}}{M^{(b)}} + \left(1 - \frac{q}{M^{(b)}}\right) \cdot \boldsymbol{\theta}_\ell^{(I)}. \tag{6}$$

**Modelling genotyping errors.** We account for genotyping error when modelling the per-sample baseline allele counts $\mathbf{C}_\ell^{(i)}$ and $\mathbf{C}_\ell^{(b, q)}$. Genotyping errors are modelled using a user-specified right-stochasic error matrix $\delta_\ell$ for each locus $\ell$, with the interpretation that $\delta_\ell(\alpha, \alpha')$ yields the probability that an allele called as $\alpha$ in a baseline sample $b$ or $i = 1, \ldots, n$ matches an allele called as $\alpha'$ in a recurrent sample $r$ at locus $\ell$. The PfRecur framework is marker-agnostic, in that the non-parametric error model $\delta_\ell$ can be adapted to different marker types.

In the present study, we consider a normalised geometric model adapted to length polymorphic microsatellite markers (akin to ref. 8), parametrised by a genotyping error probability $\varepsilon$ where, for allele lengths $i, j$

$$\delta_\ell(i, j) = \begin{cases} \frac{\varepsilon^{|i-j|+1}}{\sum_{k \neq i} \varepsilon^{|i-k|}} & \text{if } j \neq i \\ 1 - \varepsilon & \text{if } j = i \end{cases}, \tag{7}$$

where the sum in the denominator is taken over the allelic lengths in the set $H_\ell$.

**Deriving moments of allele counts.** The locus-wise probability (4) of observed genotypes for recurrent sample $r$ is formulated as a multinomial expression of the population allele frequencies $\boldsymbol{\theta}_\ell^{(S)}$. Convolving Equation (4) over the modelled distributions of $\boldsymbol{\theta}_\ell^{(I)}$ (5) and $\boldsymbol{\theta}_\ell^{(C)}$ (6), which are conditioned on per-sample allele counts, yields a multinomial expression of baseline allele counts (Supplementary Equation (12)).

Because individual clones and the alleles they carry are not directly observable in multiclonal samples genotyped using standard methods (e.g., not single-cell genotyping), allele counts for multiclonal samples must be derived under an appropriate model[16–20]. By Jensen's inequality, the expected per-sample allele counts $\mathbb{E}[\mathbf{C}_\ell^{(z)}]$ (which are straightforward to compute) cannot be substituted directly for $\mathbf{C}_\ell^{(z)}$ in Supplementary Equation (12). Convolving the locus-wise probability given by Supplementary Equation (12) over compatible allelic configurations in baseline samples $b$ and $i = 1, \ldots, n$ requires calculating moments of the per-sample allele counts

$$\mathbb{E}\left[\left(\sum_{\alpha \in A} C_\ell^{(b,q)}(\alpha)\right)^m\right] = \sum_{k=1}^{q} k^m \cdot \mathbb{P}\left(\sum_{\alpha \in A} C_\ell^{(b,q)}(\alpha) = k\right)$$

and the pooled-sample allele counts (i.e., per-sample allele counts summed over one or more baseline samples)

$$\mathbb{E}\left[\left(\sum_{i=1}^{n} \sum_{\alpha \in A} C_\ell^{(i)}(\alpha)\right)^m\right] = \sum_{k=1}^{M^{(1)}+\cdots+M^{(n)}} k^m \cdot \mathbb{P}\left(\sum_{i=1}^{n} \sum_{\alpha \in A} C_\ell^{(i)}(\alpha) = k\right)$$

aggregated over allelic subsets $A \subseteq H_\ell$ (Supplementary Note 1.2).

In Supplementary Note 1.3, we derive these moments from first principles under our framework. In brief, we use a simple combinatorial argument based on ordered partitions to derive probability mass functions and consequently cumulants of the per-sample allele counts. To adjust per-sample allele counts for genotyping error, we adopt a Poisson binomial model: the number of distinct alleles in a given set $A \subset H_\ell$ that are harboured by clones in a baseline sample is modelled as a sum of $|G_\ell|$ independent, but not identically distributed, Bernoulli random variables with respective success probabilities $\sum_{\alpha \in A} \delta_\ell(g, \alpha), g \in G_\ell^{(q)}$.

To compute moments of the pooled-sample allele counts (under the assumed independence of clones within and between baseline samples), we first sum cumulants of the per-sample allele counts to recover cumulants of the pooled-sample allele counts. We then exploit complete exponential Bell polynomials to map cumulants of the pooled-sample allele counts to moments of the pooled-sample allele counts, as required. We adopt this construction because the order of these moments (up to $M^{(r)}$) is likely, in practical settings, to be smaller than the number of baseline samples $n$ over which allele counts are aggregated.

**Prior of the statistical model**
Since mixtures of newly-inoculated and recrudescent parasite clones are comparatively unlikely in low transmission transmission settings, we take a symmetric prior, which weights pure reinfections and pure

recrudescences more heavily than intermediate mixtures. We implement this through a symmetric beta binomial distribution

$$M_C \sim \text{BetaBinomial}(M^{(r)}, \beta, \beta)$$

with $0 < \beta \leq 1$. In the case $\beta = 1$, we recover a uniform distribution over the breakdown of newly-inoculated vs recrudescent clones. In the limit $\beta \to 0$, the prior probability of all intermediate mixtures approaches zero, whereby reinfection and recrudescence constitute mutually exclusive categories.

**Posterior metrics**
We generate two posterior metrics for each recurrence: the posterior probability of at least one recrudescent clone

$$\text{M1} := 1 - \mathbb{P}\left(M_C = 0 \mid G^{(r)}, G^{(b)}, G^{(1)}, \ldots, G^{(n)}, \omega, \delta, \beta\right) \tag{8}$$

and the posterior expected proportion of recrudescent clones

$$\text{M2} := \frac{1}{M^{(r)}} \sum_{m=0}^{\min(M^{(b)}, M^{(r)})} m \cdot \mathbb{P}\left(M_C = m \mid G^{(r)}, G^{(b)}, G^{(1)}, \ldots, G^{(n)}, \omega, \delta, \beta\right). \tag{9}$$

**Application of posterior metrics**
For conceptual consistency with the CDC model[8], we perform classification using metric M1 of PfRecur: a recurrent sample $r$ is classified as a recrudescence if M1 > 0.5, and as a reinfection otherwise. Downstream efficacy estimates are computed using metric M1, rather than dichotomised classifications, in line with recommendations from the CDC[8,13]. Metric M2 serves as a supplementary descriptor for recurrences comprising a mixture of newly-inoculated and recrudescent clones, and is used to validate the model against simulated data for which these mixtures are known (see below).

**R software**
PfRecur is implemented as an R package (available at https://github.com/somyamehra/PfRecur)[39]. This largely relies on base R functionality[37], with additional dependencies on `copula::Stirling2` and `copula::Stirling1` (V1.1-1)[40] (to evaluate Stirling numbers of the second and unsigned first kind respectively); `PDQutils::cumulant2moment` and `PDQutils::moment2cumulant` (V0.1.6)[41] (to map between cumulants and moments respectively); `poisbinom::dpoisbinom` (V1.0.1)[42] (to evaluate the density function of the Poisson binomial distribution); and `VGAM::dbetabinom.ab` (V1.1-9)[43] (to evaluate the density function of the beta binomial prior). The package accommodates samples with a MOI of up to 9 (to avoid numerical instability when evaluating the posterior).

As input, the PfRecur package requires categorical (presence/absence) genotypic data in a list of binary matrices, where each matrix corresponds to a marker; named matrix columns correspond to alleles; named matrix rows correspond to samples; and matrix elements are set to 1 if the corresponding allele has been detected in the relevant sample, and 0 otherwise. Additional user-specified parameters include the per-clone marker-wise probability of detection $\omega$, and a list of marker-wise row-stochastic genotyping error matrices $\delta_\ell$.

Given a recurrent sample $r$, paired baseline sample $b$, and baseline samples $1, \ldots, n$, the function `PfRecur::evaluate_posterior` returns the discrete posterior distribution for $M_C$ over the state space $\{0, \ldots, M^{(r)}\}$ in addition to metrics M1 (8) and M2 (9), in the form of a named list.

## Simulation study

To validate PfRecur, we simulate recurrent samples as mixtures of newly-inoculated and recrudescent clones (Supplementary Note 2). In brief, we consider samples with MOIs up to 9 (mean baseline MOI ≈ 3), genotyped at 7 unlinked multi-allelic markers (each with between 10 and 30 distinct alleles). We permit siblings within samples, violating the assumed independence of clones under PfRecur. Each simulated clone is detected at each marker with probability $\omega = 0.9$. Genotyping error is applied to the set of alleles harboured by detected clones in each baseline sample with probability $\varepsilon = 0.05$, in accordance with the length-dependent normalised geometric model (7). We simulate 40 baseline datasets, each comprising 25 samples. For a baseline sample of MOI $M^{(b)}$, we simulate paired recurrences with MOI $M^{(r)} = 1, \ldots, 9$ and $m = 0, \ldots, \min\{M^{(b)}, M^{(r)}\}$ recrudescent clones. We apply our probabilistic classifier PfRecur to recover the posterior distribution for the number of newly-inoculated vs recrudescent clones within each simulated recurrent sample under a uniform prior ($\beta = 1$); the underlying parameters $\omega$ and $\varepsilon$ are assumed to be known. The results in the main text, pertaining to metrics M1 and M2, are aggregated across 29077 simulated recurrences.

## Reanalysis of[13]

Using PfRecur, we perform a re-analysis of ref. 13, with baseline samples stratified by study site (whereby allele frequencies for newly-inoculated clones are modelled to be site-specific). By default, we set $\beta = 0.25$ for the prior; $\omega = 0.9$ for the per-clone marker-wise probability of detection in each baseline/recurrent pair; and $\varepsilon = 0.05$ for the genotyping error probability, under the normalised geometric model (7). We additionally perform a sensitivity analysis for $\omega \in [0.75, 1]$ and $\varepsilon \in [0, 0.25]$. Classification is performed with both the entire 7 microsatellite marker set, and also omitting the *TA109* marker that appears to generate artefacts.

We compare posterior metric M1 of PfRecur against the gold-standard CDC model[8], which was used originally to analyse[13]. In the present study, we have re-run code provided openly by Plucinksi and colleagues[30] (with 100,000 iterations for the Gibbs sampler); posterior probabilities based on all 7 microsatellite markers may differ from those reported in[13] due to the stochastic nature of the MCMC algorithm.

## False positive recrudescence rates

To estimate false positive rates for calling recrudescence, we generate 500 artificial 'not-recrudescence' datasets from[13] by generating random derangements of baseline study participants labels within each study site, whereby permuted baseline/recurrent pairs cannot be derived from the same individual and therefore cannot represent recrudescences. The generation of permuted datasets, rather than permuted pairs, is necessitated by the construction of the CDC model[8]. We perform classification for these permuted datasets using both PfRecur (metric M1) and the CDC model[8,30] (with 10,000 iterations for the Gibbs sampler due to computational time constraints). For each permuted dataset, we compute the false positive recrudescence rate by averaging the posterior probability of recrudescence (under the CDC model) or metric M1 (under PfRecur) over recurrent samples. In addition, we consider a match-counting approach, treating the presence of one or more shared alleles at 4 or 5 markers (or more) as evidence of recrudescence. For the panel of 7 neutral microsatellites, no WHO-endorsed guidelines are available but[13,24,25] support the ≥4/7 rule. We note that current WHO guidelines, tailored to a three marker panel, stipulate a strict 3/3 match-counting rule for primary analysis[9]. We do not consider the strict 7/7 match-counting rule for the 7 neutral microsatellite panel used in[13], given that the microsatellite marker *TA109* appears to be problematic. There are also several recurrences in[13] (namely, ZL21-292, ZQ21-103 and ZL21-245) where mismatch at a single marker (with a length difference plausibly attributable to either genotyping or human error) supports the use of a relaxed match-counting rule. To avoid ambiguity, we restrict match-counting classification to baseline/recurrent pairs with at least one allele call at each of the 7 markers.

## Reporting summary

Further information on research design is available in the Nature Portfolio Reporting Summary linked to this article.

## Data availability

This study uses open access data that has previously been published by[13]. Parasite densities and clinical metadata have been retrieved from Supplemental Table S4 of ref. 13, while genotypic data have been retrieved from an accompanying GitHub repository[30].

## Code availability

The PfRecur framework has been implemented in an eponymous R package, openly available in a GitHub repository: https://github.com/somyamehra/PfRecur. The version of PfRecur (v2) used in this manuscript has been linked to Zenodo[39]: https://doi.org/10.5281/zenodo.16965130 We have re-run code provided openly by Dr Mateusz Plucinski and colleagues (with very minor input/output modifications), which implements the model detailed in[8] and is available in a GitHub repository[30]: https://github.com/MateuszPlucinski/AngolaTES2021. For completeness, all code and data relevant to this study (including the data of ref. 13 and the implementation of ref. 8) have been collated in a GitHub repository: https://github.com/somyamehra/PfTreatmentFailure.

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

## Acknowledgements

We thank Dr Mateusz Plucinski from Centres for Disease Control and Prevention, Atlanta for thoughtful and constructive criticism of the manuscript. We thank all the authors of[13] for generously making their code and data openly available. JAW is a Sir Henry Dale Fellow funded by the Wellcome Trust (223253/Z/21/Z). NJW is a Principal Research Fellow funded by the Wellcome Trust (093956/Z/10/C). A CC BY or equivalent licence is applied to the author accepted manuscript arising from this submission, in accordance with the grant's open access conditions. ART is a Marie Skłodowska-Curie Fellow (project number 101110393) funded by the European Union. Views and opinions expressed are however those of the author(s) only and do not necessarily reflect those of the European Union or granting authority. Neither the European Union nor the granting authority can be held responsible for them.

## Author contributions

Methodology: S.M. and J.A.W. Formal analysis, visualisation: S.M. Writing (original draft): S.M., A.R.T., N.J.W. and J.A.W. Writing (review and editing): S.M., A.R.T., M.I., N.J.W. and J.A.W. Conceptualisation: N.J.W. and M.I. Supervision: N.J.W. and J.A.W.

## Competing interests

The authors declare no competing interests.
