## [Peer Review file · Nature Communications]

Probabilistic classification of late treatment failure in uncomplicated falciparum malaria

Corresponding Author: Ms Somya Mehra

Version 0:

Reviewer comments:

Reviewer #1

(Remarks to the Author)

This paper reports a new statistical method for analysing molecular genotyping data on late treatment failure in uncomplicated malaria. As the authors explain, this work may have considerable importance if it prevents misinterpretation of treatment failure data. The quantitative results are of considerable practical importance.

The novelty of the paper rests on it being an advance over previous work, and the authors should be explicit about how their method differs in assumptions and/or numerical approaches from what has been done previously, in particular by the CDC group who have used similar Bayesian approaches (reference 8) but whose method clearly gives different results. While the authors provide a thorough summary of their own model, it is not clear why there are such large the differences between the new results and those of the CDC model.

The text from line 47 onwards is a very high-level summary of the issues with match-counting and gives the impression that these mostly relate to the precision of the genotyping technology. The authors should also note that issues arise with whether clones that are present in the host are present in any given blood sample, and with false-positive signals from gametocytes, dying or dead asexual parasites. Statistical model-based approaches should aim to address these issues as well as those that arise from parasite genetics. An important potential bias in naïve solutions to the problem is that not all parasite clones present in the host at any given time are present in the blood sample. These clones may be present at subsequent time points, falsely appearing to be new infections. On lines 111-112 the authors acknowledge that this may be an issue and allow for incomplete detection of clones in in baseline isolates in their simulations (but not in their model). Does the new method differ from that of CDC in how these issues are addressed?

In lines 115-120 the authors consider different thresholds for classifying clones as recrudescing.

- Is the difference between methods in the false discovery rate an artefact of the use of different thresholds?
- The population estimate of the proportion of clones that recrudescence is the quantity that should be of interest to programs, and around which it would be good to see interval estimates and estimates of bias. It can be computed as the mean of the infection-specific probabilities without specifying a threshold.

The new method is evaluated using blood samples from days 1, 2, 3, 7, 14, 21 and 28.

- Do the authors use the convention that the baseline sample counts as day 0?
- If there is the usual treatment regimen and the first dose is administered at the same time as the baseline sample is taken then it is hard to understand why there is a day 3 sample.

The genotyping was carried out on "70 pairs of baseline and recurrent isolates".

- Why is the word 'isolate' rather than 'sample' used for an individual blood sample? There was no effort to isolate specific parasites, e.g. by culturing. (Elsewhere in the paper the word 'sample' is used for 'sets of samples'. So the motivation for finding an alternative word for the individual blood sample is clear, but I find 'isolate' unsatisfactory.
- How were the 'recurrent' isolates selected? These were presumably a subset of the blood samples that were positive by microscopy on days 7, 14, 21 or 28. Or were they all the positive samples?
- Does the selection of only microscopy positive samples for typing introduce a bias?

The loci in the example dataset are known to be unlinked (line 456) so this is not an assumption, rather it is part of the

structure of the method as applied here. However, other field studies use different loci. Does this matter? Does it matter if the markers are selectively neutral?

(Remarks on code availability)

Reviewer #2

(Remarks to the Author)

****General comment****

A really neat contribution addressing a consistent problem in the field. Manuscript is nicely written and the conclusions seem well supported by the results. Ultimately, the authors show that their Bayesian Model performs similar to a previously described CDC Bayesian model when $MOI=1$, yet outperforms the CDC model when $MOI > 1$. By comparison to the authors approach, the CDC model tends to overestimate the probability of recrudescence (i.e., higher false positive rate). This was established based on a permuted dataset of 500 non-recrudescence sample pairings. Posterior probabilities of recrudescence greater than 0.5 (for both methods) are classified as a recrudescence. Among 36 recurrent infections shown in Figure 3 (from a previously published dataset), 6 <unless I am miscounting> discrepant recrudescence calls occurred across the two methods... Quite a lot.

While it is somewhat of an aside, I am interested in this observation that higher MOI values correlated to lower parasite densities. Perhaps the authors would like to comment on why this might be the case, and impacts of this on the laboratory analysis and downstream interpretation (i.e., lower parasite densities will likely reduce amplification success of markers, and if lower parasite densities occur when MOI is higher, this might further decrease the chance of detecting all/most alleles at any given marker).

****Code availability and reproducibility****

I was able to install PfRecur without issue, via:
`devtools::install_github("somyamehra/PfRecur")`

Following this, the library loaded without issue. I was also able to follow the vignettes on the authors' github page to execute the code. I cross-checked the posterior probabilities of recrudescence for several patients from the Angola study against the results reported in Figure 3 and Figure 11. The values I observed matched the reported values.

I have no major comments or criticisms. Nicely done.

(Remarks on code availability)

See above, in my responses to the author. Code seems to install easily and runs reproducibly.

Version 1:

Reviewer comments:

Reviewer #1

(Remarks to the Author)

I am happy to see that my concerns about the previous version have been comprehensively addressed.

(Remarks on code availability)

Responses to reviewers' comments: manuscript NCOMMS-25-07344

Probabilistic classification of late treatment failure in uncomplicated malaria

Somya Mehra, Aimee R Taylor, Mallika Imwong, Nicholas J White, James A Watson

July 8, 2025

Reviewer #1 (Remarks to the Author):

This paper reports a new statistical method for analysing molecular genotyping data on late treatment failure in uncomplicated malaria. As the authors explain, this work may have considerable importance if it prevents misinterpretation of treatment failure data. The quantitative results are of considerable practical importance.

The novelty of the paper rests on it being an advance over previous work, and the authors should be explicit about how their method differs in assumptions and/or numerical approaches from what has been done previously, in particular by the CDC group who have used similar Bayesian approaches (reference 8) but whose method clearly gives different results. While the authors provide a thorough summary of their own model, it is not clear why there are such large the differences between the new results and those of the CDC model.

We thank the reviewer for their comments. We would argue that this model is more than just an advance on previous work (the key similarity is that it uses a Bayesian framework). It provides a completely novel framework for calculating the probability of recrudescence for paired falciparum samples. We have added a sentence to the main text highlighting Appendix D which outlines the core differences between the models (Appendix D was included, but not mentioned in the main text of the original submission; we apologise for the oversight).

Lines 106 to 107: “An extended comparison of the *PfRecur* model structure against the Bayesian CDC model [6] is provided in Appendix D.”

Lines 1006 to 1011, Appendix D, original submission: “In brief, the CDC model [6] explicitly estimates allelic configurations using a Gibbs sampler; averages the likelihood of recrudescence across pairs of clones in a baseline and recurrent isolate; and concurrently generates estimates of parametric (geometric) genotyping error probabilities and a per-clone unobserved allele “penalty”. In contrast, in *PfRecur* we average analytically over compatible allelic configurations in each isolate to obtain an analytically tractable posterior; model recurrent isolates as mixtures of newly-inoculated vs recrudescence clones; and treat both genotyping error and the per-clone marker-wise probability of detection for paired baseline/recurrent isolates as user-specified parameters.”

The Discussion outlines the core issues with the CDC model which result in pathological behaviour.

Lines 206 to 227 (original submission): “The Bayesian model proposed by the CDC has three potential issues. Firstly, a per-clone unobserved allele “penalty” — a multiplicative factor applied to the likelihood when the imputed allele for a clone lies outside the observed set of alleles for a given isolate — is estimated simultaneously during classification (a relatively lax penalty in the range 0.25 to 0.45 was estimated for each study site in [8]). We suggest that this construction increases the estimated probability of recrudescence. This is because the unobserved alleles have little effect on the probability of reinfection (they only affect allele frequency estimates), but tend to augment the chance of allelic matches between the paired isolates and consequently the probability of classifying as a recrudescence. This bias is particularly pronounced for isolates with no genotyping data at a subset of markers, and isolates with unbalanced cardinality across loci. Markers with missing genotypes are included in the estimation procedure, and the per-clone unobserved allele penalty is applied irrespective of whether or not imputed alleles match those in the paired isolate. Isolates with unbalanced cardinality across loci result in a greater chance of introducing unobserved alleles, a feature that may be problematic when MOIs are inflated due to artificially high multiplicities at one marker (e.g. TA109 in the dataset analysed here). Secondly, estimation of the genotyping error appears to be unstable. In the dataset analysed, site specific estimates of genotyping error varied from 0.025 in Lunda Sul to 0.072 in Benguela (i.e. a three fold difference). However, all genotyping was presumably done by the same central laboratory. The difference in estimates suggests possible identifiability issues in the model. As demonstrated by the marker TA109, a major determinant of genotyping error is likely to be the parasite density in the filter paper. Thirdly, the reliability of the output model probabilities depend on convergence of the algorithm. To the best of our knowledge, there has been no in depth study of convergence of this algorithm, and the default parameters suggested (1000 iterations) appear insufficient.”

We have now added additional exposition in the Discussion to highlight key differences between our model and the CDC model.

Lines 234 to 241: “In contrast, *PfRecur* averages over compatible allelic configurations within each sample — accounting for the imperfect detection of clones in the paired baseline and recurrent samples under a marker-wise truncated binomial model governed by the user-specified per-clone probability of detection ω , and user-specified marker-wise non-parametric genotyping error matrices — to directly evaluate an analytically-tractable (discrete) posterior distribution for the number of newly-inoculated vs recrudescing clones within each recurrent sample, allowing for classification of recurrent samples with multiplicity of infection up to 9 in the order of seconds.”

The text from line 47 onwards is a very high-level summary of the issues with match-counting and gives the impression that these mostly relate to the precision of the genotyping technology. The authors should also note that issues arise with whether clones that are present in the host are present in any given blood sample, and with false-positive signals from gametocytes, dying or dead asexual parasites. Statistical model-based approaches should aim to address these issues as well as those that arise from parasite genetics. An important potential bias in naïve solutions to the problem is that not all parasite clones present in the host at any given time are present in the blood sample. These clones may be present at subsequent time points, falsely appearing to be new infections. On lines 111-112 the authors acknowledge that this may be an issue and allow for incomplete detection of clones in baseline isolates in their simulations (but not in their model). Does the new method differ from that of CDC in how these issues are addressed?

We have rephrased this section of the Introduction and added several sentences in the Discussion to focus on the issues that model-based approaches can resolve. There is a limit to what these models should be able to do. Our view is that if there is a clone that has not been detected at time of treatment but which appears at time of recrudescence then the model should call this a reinfection.

Lines 49 to 53 (Introduction): “Model-based approaches for molecular correction can address classification uncertainty, take into account background allele frequencies, and adjust for multiple comparisons. In our view, additional biological complexities (for example, the presence of an asynchronous sequestered clone, or amplification of residual gametocyte DNA post-treatment [1]) constitute a distinct problem. These issues are beyond the scope of a general-purpose statistical model for molecular correction.”

Lines 201 to 205 (Discussion): “The statistical model cannot solve all technical and biological complexities, but simply provide a principled framework for handling paired genotypic data from polyclonal infections. Possible complexities which would confound interpretation of outcomes such as an asynchronous clone, sequestered at time of treatment and causing a later recrudescence infection [1], are out of scope for a general purpose statistical model.”

A modular approach, combining molecular correction with time series models of the blood-sampling process [3] may address the complexities raised by the reviewer. For instance, while long term persistence of gametocytes is possible [5], antimalarial treatment, particularly with artemisinin derivatives [7], will accelerate gametocyte clearance. Sufficiently high baseline gametocytemia may mean that gametocytes are detected in the recurrent infection, but this may be better addressed by a supplementary model with baseline parasitemia and gametocytemia treated as covariate.

Additionally, we would expect major confounders that are not specifically related to drug resistance to lead to low cure rates across all study arms and sites, contrary to the high cure rates for some sites and drug regimens seen in [8] and other studies. Residual intraerythrocytic DNA or DNA derived from dead/dying asexual parasites may be detectable for several weeks, and may therefore be a confounder for early recurrence; but if this is a major confounder, then apparent cure rates should be low for all study sites. Similarly, as resistance increases, recrudescence misclassified as reinfection because of the sequestration of asynchronous clones should increase, whereby apparent cure rates should be low across all sites. If this phenomenon occurs in a low transmission setting with monoclonal infections, then this should lead to slide-negative malaria which is yet to be proven.

We wish to clarify that our model accounts for incomplete detection of clones in the baseline sample paired to the recurrent sample of interest, but not the remaining baseline samples from which allele frequencies are derived. The Discussion outlines the differences in how missing/undetected alleles are dealt with by the two methods, as addressed above. A detailed comparison of how the incomplete detection of clones is handled in our model vs the CDC model is provided in Appendix D.

Selected rows of the Supplementary Table in Appendix D:

Feature	CDC model [6]	PfRecur
---------	---------------	----------------

Penalising unobserved alleles	Number of hidden alleles across all isolates defined to be the difference between MOI and the locus cardinality; implement a per-clone unobserved allele penalty q whenever a hidden allele lies outside the set of observed alleles; q estimated under a beta-binomial model in the Gibbs sampler	Implement a marker-wise probability ω of detecting each clone in the paired baseline and recurrent isolates; ω is user-specified
Adjusting for unobserved alleles	Update one randomly chosen hidden allele in a baseline/recurrent pair per iteration; sample hidden alleles uniformly at random over the set of possible alleles at that marker; likelihood ratio for Gibbs sampler based on the convolution over frequency of each allele (true underlying allele) and genotyping error (probability true allele appears as sampled hidden allele), multiplied by the per-clone unobserved allele penalty q if relevant; if a switch is to be made, add a length variability term to sampled hidden allele (adjusted for allelic binning: based on the mean SD within allelic classes at the marker)	Impute allelic states for undetected clones in the paired baseline isolate based on derived population allele frequencies for newly-inoculated clones, with a marker-wise truncated binomial model for the number of clones detected at each marker; derive the locus-wise likelihood of observed genotypes for the recurrent isolate over detected clones only, with a truncated multinomial model for the number of newly-inoculated vs recrudescence clones detected at each marker

In lines 115-120 the authors consider different thresholds for classifying clones as recrudescence.

- *Is the difference between methods in the false discovery rate an artefact of the use of different thresholds?*

No, the CDC method has an inherently higher false discovery rate when the paired samples have $MOI > 1$; please refer to Figure 3 and the sensitivity analyses in Appendix C of the original submission, which do not rely on any thresholds. For recurrences in [8], a comparison of the posterior probability of recrudescence under the CDC model vs metric M1 (the probability of at least one recrudescence clone) under *PfRecur* (with default parameters) is shown in Figure 3 when the baseline $MOI > 1$; recurrences with differing classifications are highlighted in Supplementary Figure 5 (Figure 11 of the original submission). Results under the CDC model vs *PfRecur* for the subset of paired samples which are sensitive to the user-defined parameters under *PfRecur* (namely, the marker-wise per-clone detection probability ω and genotyping error probability ϵ) are shown in Supplementary Figures 1 and 2 (Figures 7 and 8 of the original submission) respectively. A comparison of false discovery rates under the CDC model vs *PfRecur* with different per-clone detection probabilities ω over 500 permuted artificial ‘not-recrudescence’ datasets generated from [8] is shown in Supplementary Figures 3 and 4 (Figures 9 and 10 of the original submission); these false discovery rates have been calculated for each permuted dataset by averaging the posterior probability of recrudescence for the CDC model vs metric M1 under our model, without imposing any thresholds.

- *The population estimate of the proportion of clones that recrudescence is the quantity that should be of interest to programs, and around which it would be good to see interval estimates and estimates of bias. It can be computed as the mean of the infection-specific probabilities without specifying a threshold.*

We have not reported efficacy estimates derived from simulated data, because our simulation model does not account for transmission dynamics, drug kinetics or loss to follow up. The primary purpose of our simulation model is to evaluate the per-recurrence performance of our probabilistic classifier given several forms of model misspecification.

We instead report efficacy estimates from our re-analysis of Dimbu et al. [8] in Supplementary Table 3 (Table 4 in the original submission) of Appendix C.3.1. In accordance with previously-published analyses [6, 8, 9], efficacy is calculated by embedding model-derived posterior probabilities of recrudescence into a Kaplan-Meier survival model accounting for right censoring and loss to follow-up. For comparability to previous analyses, we embed metric M1 (the posterior probability of at least one recrudescence clone) rather than metric M2 (the posterior proportion of recrudescence clones) of our model in this Kaplan-Meier correction.

Lines 992 to 998, original submission: “To generate efficacy estimates adjusted for classification uncertainty and right censoring, we perform the Kaplan-Meier correction used in Dimbu et al. [8]¹: for each study arm, we perform 10,000 iterations of bootstrap resampling whereby we sample a binary reinfection or recrudescence state for each patient under a Bernoulli distribution with success parameter given by the posterior probability of recrudescence (or metric M1 under *PfRecur*); and then compute Kaplan-Meier estimates (including the lower 2.5% and upper 97.5% confidence interval under the `peto` method implemented in the R function `survival::survfit` [2]) for these binary infection states.”

These efficacy estimates are referenced in the Discussion.

Lines 200 to 205, original submission: “When applied to microsatellite data from a recent therapeutic efficacy assessment in Angola [8], our classification model outputs substantially different estimates for one third of the recurrent infections with baseline MOI > 1 [6]; however this does not greatly impact efficacy estimates across study arms (Supplementary Table 4). The systematic overestimation of recrudescence rates is likely to be greater in settings with greater parasite diversity, higher polyclonality, and more frequent reinfection.”

Averaging the proportion of recrudescence clones (metric M2) across recurrent infections ($\frac{1}{N} \sum_{i=1}^N M2_i$) is an alternative measure of efficacy. Without a time-to-event prior, it does not account for censoring and loss to follow up. We don’t report it because it is not comparable with previously published analyses.

The new method is evaluated using blood samples from days 1, 2, 3, 7, 14, 21 and 28.

- *Do the authors use the convention that the baseline sample counts as day 0?*

In accordance with the terminology used by the original study authors [8], enrolment is designated day 0; we have now clarified this point in the main text.

Lines 269 to 272: “Follow-up, entailing clinical examination and slide microscopy, occurred on days 1 (clinical examination only), 2, 3, 7, 14, 21 and 28 in addition to days 35 and 42 for patients treated with dihydroartemisinin-piperaquine and artesunate-pyronaridine, with the convention that enrolment (baseline) was designated day 0.”

We wish to clarify that the new method has been applied to data from late treatment failures, and thus a subset of blood samples from day 7 onwards, in addition to genotyped day 0 samples.

- *If there is the usual treatment regimen and the first dose is administered at the same time as the baseline sample is taken then it is hard to understand why there is a day 3 sample.*

We thank the reviewer for these questions regarding the study conduct. We stress that we are re-using available data from a previous treatment efficacy study, conducted by Dimbu et al. [8]. The focus

¹Implemented in <https://github.com/MateuszPlucinski/AngolaTES2021> [9]

of our paper is the statistical methodology for the correct analysis of these data and not the trial or laboratory methodology. However, we note that each drug regimen evaluated in the study [8] was administered over 3 days. According to a protocol published by the World Health Organisation (WHO) [4], on which the study [8] is based, “prevalence of patients who are positive on day 3 is used as an indirect clinical marker of artesunate resistance on the Thai–Cambodian border”.

The genotyping was carried out on “70 pairs of baseline and recurrent isolates”.

- *Why is the word ‘isolate’ rather than ‘sample’ used for an individual blood sample? There was no effort to isolate specific parasites, e.g. by culturing. (Elsewhere in the paper the word ‘sample’ is used for ‘sets of samples’. So the motivation for finding an alternative word for the individual blood sample is clear, but I find ‘isolate’ unsatisfactory.*

We have changed ‘isolate’ to ‘sample’ throughout the text.

- *How were the ‘recurrent’ isolates selected? These were presumably a subset of the blood samples that were positive by microscopy on days 7, 14, 21 or 28. Or were they all the positive samples?*

Dimbu et al. [8] identified a total of 71 late treatment failures, characterised by the “presence of any *P. falciparum* asexual parasites on slide microscopy on or after day 7”. Genotyping was performed for 70 of these 71 late treatment failures (one recurrent sample from Benguela was deemed missing) [8].

- *Does the selection of only microscopy positive samples for typing introduce a bias?*

Yes, sub-microscopic infections could bias results if the probability of being sub-microscopic is not equal for recrudescence and reinfection, but this is a limitation of the study design. The study performed by Dimbu et al. [8] was informed by a protocol developed by WHO for antimalarial therapeutic efficacy studies [4], which posits microscopy-detectable *P. falciparum* parasitemia as an inclusion criterion for enrolment, and for the ascertainment of recurrent infection. We reiterate that the focus of our paper is the analysis of genotyping data from therapeutic efficacy studies, not field methodology.

Under our model, classification is performed separately for each recurrent infection. Allele frequencies for newly-inoculated clones in the recurrent infection, however, are derived from all available genotyped baseline samples from the relevant site, i.e., a subset of microscopy positive infections at enrolment; these may be enriched for clones against which limited immunity has been acquired by individuals in the population. Randomisation of study arms and the baseline samples selected for genotyping may mitigate biases in the estimation of allele frequencies for newly-inoculated clones; however, we would argue that this is principally a question of study design/methodology, rather than data analysis.

The loci in the example dataset are known to be unlinked (line 456) so this is not an assumption, rather it is part of the structure of the method as applied here. However, other field studies use different loci. Does this matter? Does it matter if the markers are selectively neutral?

We would argue that unlinked loci constitute a model assumption that is satisfied by the example dataset. To the best of our knowledge, the length polymorphic and microsatellite markers that are typically used for molecular correction (namely, *MSP1*, *MSP2* and *GLURP*, in addition to the 7 microsatellite panel used in [8]) are known to be unlinked. A panel of linked markers would be less informative than a panel of unlinked markers for distinguishing non-clones from clones, making less conceptual sense from the perspective

of molecular correction algorithms predicated on allele-matching.

In generality, neutral markers on different chromosomes would be expected to be unlinked, given random assortment of chromosomes in meiosis. As an example to the contrary, non-neutral *Pfdhfr* and *Pfdhps* markers are on different chromosomes, but in linkage disequilibrium due to drug pressure and the involvement of the same mechanism. If markers are not selectively neutral, there may be systematic patterns that could be exploited under a model-based approach: for instance, resistance-associated alleles would be enriched in recrudescence clones and would therefore be informative in identifying recrudescence.

Reviewer #2 (Remarks to the Author):

General comment

A really neat contribution addressing a consistent problem in the field. Manuscript is nicely written and the conclusions seem well supported by the results. Ultimately, the authors show that their Bayesian Model performs similar to a previously described CDC Bayesian model when $MOI=1$, yet outperforms the CDC model when $MOI > 1$. By comparison to the authors approach, the CDC model tends to overestimate the probability of recrudescence infections (i.e., higher false positive rate). This was established based on a permuted dataset of 500 non-recrudescence sample pairings. Posterior probabilities of recrudescence greater than 0.5 (for both methods) are classified as a recrudescence. Among 36 recurrent infections shown in Figure 3 (from a previously published dataset), 6 discrepant recrudescence calls occurred across the two methods... Quite a lot.

We thank the reviewer for their kind words.

While it is somewhat of an aside, I am interested in this observation that higher MOI values correlated to lower parasite densities. Perhaps the authors would like to comment on why this might be the case, and impacts of this on the laboratory analysis and downstream interpretation (i.e., lower parasite densities will likely reduce amplification success of markers, and if lower parasite densities occur when MOI is higher, this might further decrease the chance of detecting all/most alleles at any given marker).

We have been collaborating with the Molecular Genotyping Laboratory at MORU, lead by our co-author Prof Mallika Imwong, to investigate this. Unpublished data suggests that this trend may be driven by non-specific peaks, potentially attributable to human DNA, in the marker *TA109* that become increasingly pronounced at low parasite densities. Given current model-based and match counting methods accommodate undetected clones, but do not explicitly adjust for spurious alleles, inclusion of the marker *TA109* may bias estimated recrudescence rates.

There may, however, be an additional causal pathway at play: individuals with higher acquired immunity may have lower density infections with greater intra-host diversity and thus higher MOIs. In turn, as the reviewer suggests, this may lead to a lower probability of detecting all/most alleles at any given marker for samples with higher MOIs; however, accommodating an overly lax probability of per-clone detection in a model-based method could bias recrudescence rates if spurious alleles are present, as seems to be the case for the marker *TA109*.

Code availability and reproducibility

*I was able to install PfRecur without issue, via: devtools::install_github("somyamehraPfRecur")
Following this, the library loaded without issue. I was also able to follow the vignettes on the authors' github page to execute the code. I cross-checked the posterior probabilities of recrudescence for several patients from the Angola study against the results reported in Figure 3 and Figure 11. The values I observed matched the reported values.*

I have no major comments or criticisms. Nicely done.

We thank the reviewer for their thoroughness in verifying our code.

Reviewer #2 (Remarks on code availability):

See above, in my responses to the author. Code seems to install easily and runs reproducibly.

References

- [1] G Snounou and HP Beck. "The use of PCR genotyping in the assessment of recrudescence or reinfection after antimalarial drug treatment". In: *Parasitology Today* 14.11 (1998), pp. 462–467.
- [2] Terry M. Therneau and Patricia M. Grambsch. *Modeling Survival Data: Extending the Cox Model*. New York: Springer, 2000. ISBN: 0-387-98784-3.
- [3] Tom Smith and Penelope Vounatsou. "Estimation of infection and recovery rates for highly polymorphic parasites when detectability is imperfect, using hidden Markov models". In: *Statistics in Medicine* 22.10 (2003), pp. 1709–1724.
- [4] WHO. *Methods for Surveillance of Antimalarial Drug Efficacy*. World Health Organization, 2009.
- [5] Teun Bousema and Chris Drakeley. "Epidemiology and infectivity of Plasmodium falciparum and Plasmodium vivax gametocytes in relation to malaria control and elimination". In: *Clinical Microbiology Reviews* 24.2 (2011), pp. 377–410.
- [6] Mateusz M Plucinski, Lindsay Morton, Mary Bushman, Pedro Rafael Dimbu, and Venkatachalam Udhayakumar. "Robust algorithm for systematic classification of malaria late treatment failures as recrudescence or reinfection using microsatellite genotyping". In: *Antimicrobial Agents and Chemotherapy* 59.10 (2015), pp. 6096–6100.
- [7] WWARN Gametocyte Study Group. "Gametocyte carriage in uncomplicated Plasmodium falciparum malaria following treatment with artemisinin combination therapy: a systematic review and meta-analysis of individual patient data". In: *BMC Medicine* 14 (2016), pp. 1–18.
- [8] Pedro Rafael Dimbu et al. "Therapeutic response to four artemisinin-based combination therapies in Angola, 2021". In: *Antimicrobial Agents and Chemotherapy* (2024), e01525–23.
- [9] Mateusz Plucinski. *AngolaTES2021*. <https://github.com/MateuszPlucinski/AngolaTES2021>. 2024.